# Spatiotemporal Patterns and Drivers of the Carbon Budget in the Yangtze River Delta Region, China

Qi Fu [1,2,3,†] , Mengfan Gao [1,†], Yue Wang [1] , Tinghui Wang [1], Xu Bi [4,*] and Jinhua Chen [1,2,3,*]

1    School of Politics and Public Administration, Soochow University, Suzhou 215123, China;
     fuqi@suda.edu.cn (Q.F.); 20194202061@stu.suda.edu.cn (M.G.); 20204202067@stu.suda.edu.cn (Y.W.);
     20204002013@stu.suda.edu.cn (T.W.)
2    The Institute of Regional Governance, Soochow University, Suzhou 215123, China
3    Research Institute of Metropolitan Development of China, Soochow University, Suzhou 215123, China
4    College of Resources and Environment, Shanxi University of Finance and Economics, Taiyuan 030006, China
*    Correspondence: bixu@mail.bnu.edu.cn (X.B.); jhchen@suda.edu.cn (J.C.)
†    These authors contributed equally to this work.

**Abstract:** Improving our understanding of the patterns and drivers of regional carbon budgets is critical to mitigating climate change regionally and globally. Different from previous research, our study attempts to reveal the comprehensive impact of climate change and human activities factors on the carbon budget. Based on the Carnegie–Ames–Stanford approach (CASA) model, the IPCC inventory method, the ordinary least squares (OLS) regression model, the Geodetector model, and the geographically weighted regression (GWR) method, we investigated the spatiotemporal patterns of the carbon budget in the Yangtze River Delta (YRD) region from 2000 to 2015 and analyzed the effects of climate change and human activities on the carbon budget. The results showed that the carbon budget in the YRD region changed from 271.33 million tons in 2000 to −1193.76 million tons in 2015. During this period, the changes in the carbon budget per unit area in the four provinces all showed a decreasing trend, among which Shanghai decreased the most, followed by Jiangsu, Zhejiang and Anhui. In terms of spatial pattern, the carbon budget of the YRD region has a "core-edge" structural feature. The closer it is to Shanghai, the core area, the more severe the carbon budget deficit; the farther from it, the greater the carbon budget surplus. Overall, we found that human activities have a greater impact on the carbon budget than climate change. The top three drivers were, in order, changes in population density, GDP per capita, and unused land, with q values of 0.3317, 0.1202, and 0.0998, respectively. Locally, the impact of the drivers on the carbon budget shows obvious spatial heterogeneity. In particular, the population density was negatively correlated with carbon budget changes in the entire study area, and the coefficients of GDP per capita and unused land were negative in most counties. Based on the results, we put forward suggestions for restricting population flow among the core area and the peripheral area, promoting industrial innovation in the core area and ecological protection in the peripheral area, as well as implementing three-dimensional space development in the core area and controlling the expansion of construction land in the peripheral area. Our study can provide a scientific basis for low-carbon development in the YRD region. The methodology and findings of this study can provide references for similar studies in other urbanized regions around the world.

**Keywords:** carbon budget; spatiotemporal patterns; drivers; Yangtze River Delta

## 1. Introduction

Since the industrial revolution, the global economy has developed rapidly, and various energy sources have been widely used. The resulting emissions of greenhouse gases (including $CO_2$, $CH_4$, and $N_2O$), especially $CO_2$, have led to the gradual acceleration of the rise in the global average temperature over the past 200 years [1–3]. The need to mitigate climate change has become a global consensus [4]. Terrestrial ecosystems have a strong

carbon sequestration function, and terrestrial vegetation can convert and store atmospheric $CO_2$ into organic matter through photosynthesis, playing a key role in mitigating global warming [5]. Many studies have demonstrated that carbon sequestration in terrestrial ecosystems can offset substantial anthropogenic carbon emissions [6–8]. Therefore, it is critical to uncover the relationship between carbon emissions and carbon sequestration and the drivers behind them.

The carbon budget of terrestrial ecosystems refers to the difference between carbon sequestration and carbon emissions [9]. Conducting carbon budget research can help clarify the regional carbon emission reduction pressure and carbon sink potential [10–12]. Preliminary research on regional carbon budgets mainly focused on natural ecosystems, such as forests, grasslands, farmlands, and wetlands [13–16]. In recent years, with the deepening of global change research and the proposal of a low-carbon economy, carbon budget research covering natural and human social and economic activities has begun to attract academic attention [9,17]. The assessment methods of the carbon budget mainly include field surveys, empirical models, remote-sensing models, IPCC inventory methods, etc. [18]. Field surveys can be used to obtain measured data on vegetation and soil carbon densities or the net carbon exchange between ecosystems and the atmosphere. This method is suitable for studies at the scale of sample sites and ecosystems [19]. At larger spatial scales, statistical models are usually established to evaluate carbon budgets based on empirical relationships, and such studies are typified by Houghton's bookkeeping model [20]. In recent decades, with the advancement of remote sensing and GIS technology, remote-sensing models (such as the CASA model) have become an effective technical means to assess carbon sequestration [21]. The IPCC inventory method is widely used in carbon emission accounting due to its simple calculation and high practicability [22]. Combining remote-sensing models with the IPCC inventory approach can greatly improve the efficiency and accuracy of carbon budget assessments [23]. To date, researchers have carried out many carbon budget studies based on these methods in different regions of the world [21,24,25]. However, how to apply scientific research results to guide carbon management in practice is still a challenge [26].

Before developing carbon management strategies, policymakers should understand what underlying factors affect carbon budgets and where and when these impacts occur [27,28]. Climate change and human activities are considered the two main driving factors of carbon budget dynamics [21,29–32]. Climatic change affects the ecosystem carbon budget mainly by changing vegetation phenology, photosynthesis, respiration, soil moisture and evapotranspiration [10,33]. Human activities, especially the impacts of land use/land cover change (LUCC) on carbon budgets, are important factors leading to the current increase in atmospheric $CO_2$ concentrations [34,35]. Studies by Houghton et al. showed that from 1850 to 1990, global LUCC led to the emission of 124 Pg C into the atmosphere, of which 108 Pg C came from the reduction in forest ecosystem area [20]. In addition, some researchers have focused on other indicators of human activity, such as the impact of population growth and GDP growth on the carbon budget [9,36,37]. However, most of the current driving analysis studies only focus on one of climate change [38,39] or human activities [40,41], and comprehensive analysis of carbon budget changes caused by multiple drivers is rare.

With the rapid development of the national economy, China has become a major $CO_2$ emitter [42]. To mitigate global warming, China has made unremitting efforts to increase forest and grass areas and reduce energy consumption in recent years [43]. The Chinese government has pledged to peak carbon emissions by 2030 and be carbon-neutral by 2060 [44]. Located in East China, the Yangtze River Delta (YRD) region is the largest urban agglomeration and is one of the regions with the fastest urbanization [45]. In the past few decades, energy has played an important role in promoting the rapid development of the YRD region, but it has also caused a high level of carbon emissions [46]. The unbalanced regional development in this area has become increasingly prominent, and it is urgent to formulate differentiated regional development policies. To date, some researchers have carried out research on the carbon budget at the national scale in China [47]. Other researchers have conducted many studies on either carbon sequestration or carbon

emissions in the YRD region [22,45,48–50]. However, studies on the spatiotemporal patterns of the carbon budget and its drivers in this region have not yet been performed.

Different from previous research, our study tried to reveal the comprehensive impact of human activities and climate change factors on the carbon budget and to narrow the gaps in carbon budget research in the YRD region. We estimated the carbon budgets of 308 county-level administrative units in the YRD region from 2000 to 2015 and explored the drivers of the spatiotemporal patterns of the carbon budget. The purpose of our study was to find scientific carbon management pathways in the YRD region. Given this challenge, three questions were addressed: (1) Is there a certain characteristic in the spatiotemporal pattern of the carbon budget in the YRD region? (2) Which drivers have a greater impact on the carbon budget and where do these impacts occur? (3) How should we carry out carbon management to reduce the imbalance of regional carbon budget? Our results provide a scientific basis for the low-carbon and sustainable development in the YRD region, and our methodology provides a reference for other rapidly urbanizing regions in the world.

## 2. Materials and Methods

### 2.1. Study Area

The YRD region includes Shanghai, Jiangsu, Zhejiang, and Anhui provinces (Figure 1), comprising 41 cities and 308 county-level administrative units (including counties and districts, which are uniformly expressed as counties in our research). The study area covers an area of 358,000 square kilometers, accounting for approximately 3.7% of China's total area. The YRD region is located in the lower reaches of the Yangtze River (114°54′–123°10′ E and 27°02′–35°08′ N). The region has abundant rainfall, with an annual precipitation of 704–2000 mm and an annual average temperature of 12.2–18.9 °C. The water system is well-developed and the vegetation is mainly subtropical evergreen broad-leaved forest.

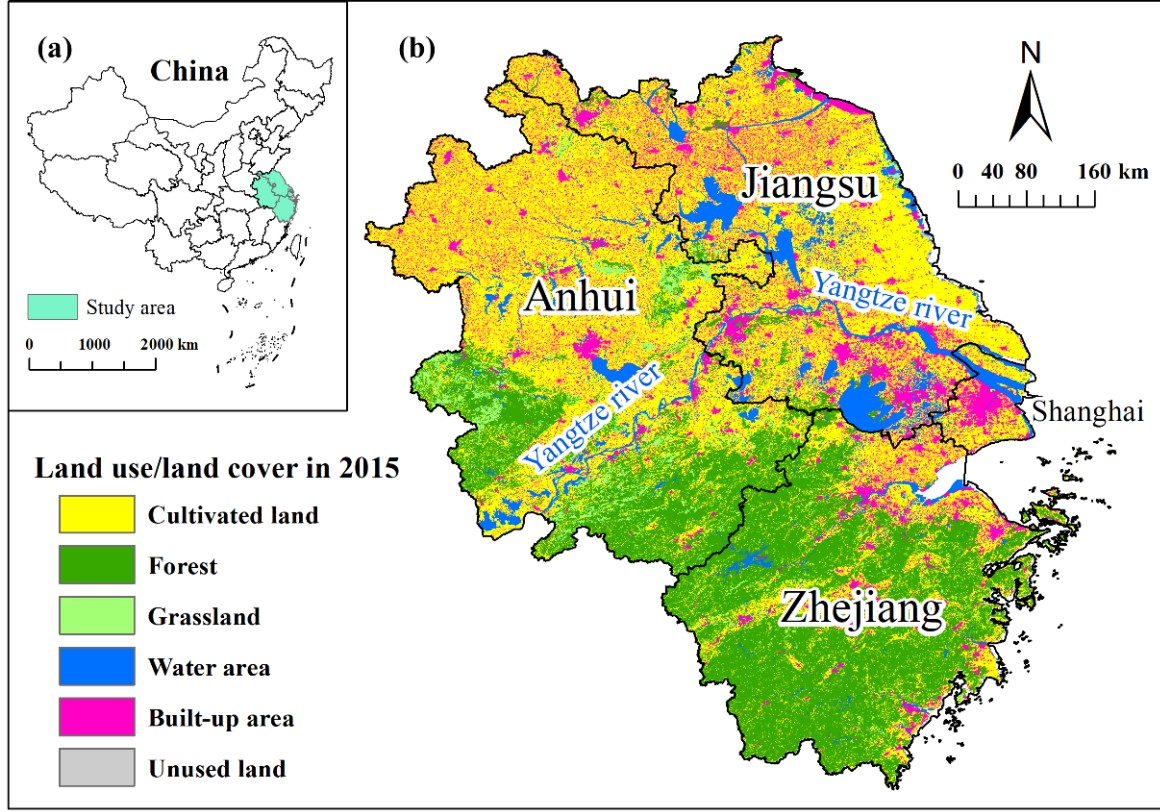

**Figure 1.** (**a**) The location of the YRD region in China; (**b**) the land use/land cover (LULC) pattern of the YRD region in 2015.

The YRD is one of the most densely populated areas in China, and it has the fastest urbanization process in the entire country [45]. During 2000 and 2015, the process of urban expansion led to a significant decline in ecosystem carbon sequestration [51]. However, carbon emissions in the YRD region continued to increase during this period. Researchers confirmed that the coupling coordination between carbon sequestration and carbon emission in this region was in an imbalanced state [52]. As of 2015, carbon sequestration had offset less than 5% of carbon emissions in eight core cities in the YRD region [53].

*2.2. Data Sources*

The data required to assess the carbon sequestration, carbon emissions, and drivers are mainly meteorological data, land use data, normalized difference vegetation index (NDVI), population density data, gross domestic product (GDP) per capita data, administrative boundary data, nighttime light data, and energy consumption data (Table 1).

**Table 1.** Data used in this study.

| Data Name | Data Type | Year | Source |
|---|---|---|---|
| County administrative boundaries in the YRD region | Vector data | 2015 | Resource environment data cloud platform (http://www.resdc.cn/Default.aspx accessed on 30 May 2021) |
| Monthly total precipitation | Site data | 2000, 2015 | China Meteorological Data Network (http://data.cma.cn/ accessed on 5 June 2021) |
| Average temperature | Site data | 2000, 2015 | China Meteorological Data Network (http://data.cma.cn/ accessed on 5 June 2021) |
| Solar radiation | Site data | 2000, 2015 | China Meteorological Data Network (http://data.cma.cn/ accessed on 15 June 2021) |
| Normalized difference vegetation index | 1 km × 1 km Raster data | 2000, 2015 | Resource environment data cloud platform (http://www.resdc.cn/Default.aspx accessed on 2 July 2021) |
| Land use/land cover | 30 m × 30 m Raster data | 2000, 2015 | Resource environment data cloud platform (http://www.resdc.cn/Default.aspx accessed on 5 July 2021) |
| Population density | 1 km × 1 km Raster data | 2000, 2015 | Resource environment data cloud platform (http://www.resdc.cn/Default.aspx accessed on 5 July 2021) |
| GDP per capita | 1 km × 1 km Raster data | 2000, 2015 | Resource environment data cloud platform (http://www.resdc.cn/Default.aspx accessed on 5 July 2021) |
| Energy emission data | Text data | 2000, 2015 | Statistical yearbook of CNKI (https://data.cnki.net/Yearbook accessed on 6 August 2021) |
| Night lights data | 1 km × 1 km Raster data | 2000, 2015 | Harvard University Database (https://dataverse.harvard.edu/dataset.xhtml?persistentId=doi:10.7910/DVN/YGIVCD accessed on 8 October 2021) |

*2.3. Methods*

The analytical workflow for this study consisted of three key steps. First, we used the CASA model, the IPCC inventory method and the PSO-BP neural network model to evaluate the carbon sequestration and carbon emissions in the Yangtze River Delta in 2000 and 2015, and then calculated the carbon budget. Second, we investigated the spatiotemporal pattern of the carbon budget based on statistical analysis and GIS spatial analysis. Third, we used the OLS model to eliminate the variables with multicollinearity from multiple potential driving factors and analyzed the driving forces from the global and local perspectives based on the Geodetector model and the GWR model, respectively. The flow chart of this study is shown in Figure 2.

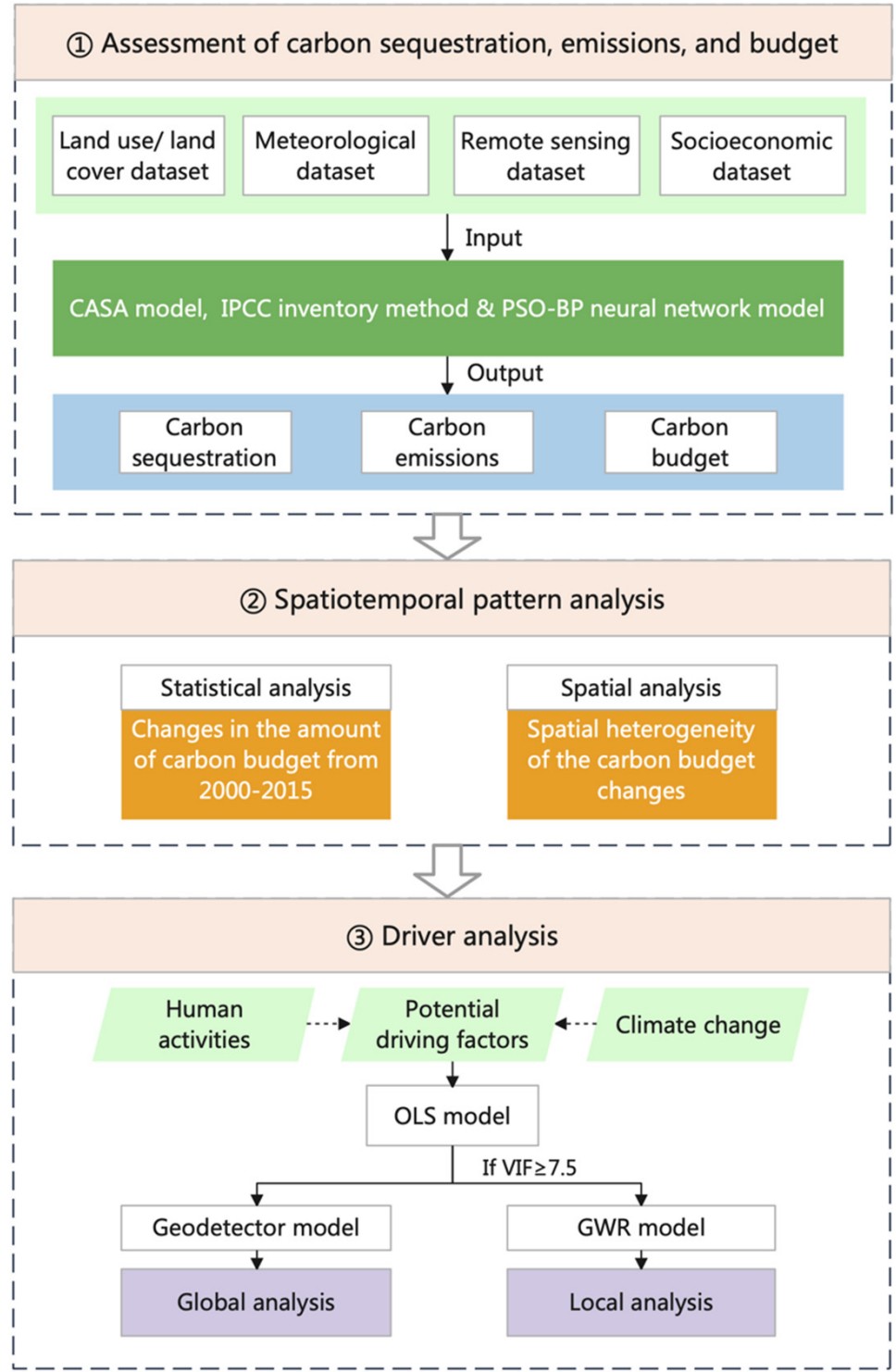

**Figure 2.** The flow chart of this study.

2.3.1. Assessment of Carbon Sequestration, Emissions, and Budget

Carbon Sequestration

The CASA model was used to estimate the *NPP* in the YRD region [54], and then, the amount of carbon sequestration in vegetation was obtained. The *NPP* estimated in the

model can be determined from the photosynthetically active radiation absorbed by the plants ($APAR$) and the actual photosynthetic efficiency ($\varepsilon$):

$$NPP(x,t) = APAR(x,t) \times \varepsilon(x,t) \tag{1}$$

In Equation (1), $APAR(x,t)$ is the photosynthetically active radiation absorbed by pixel $x$ in month $t$ (MJ·m$^{-2}$), and $\varepsilon(x,t)$ is the actual light energy utilization rate of pixel $x$ in month $t$ (MJ·m$^{-2}$). For the specific calculation, we referred to the research of Yang et al. [55].

According to vegetation photosynthesis, every 1 g of dry matter produced by the ecosystem can absorb 1.62 g of $CO_2$, and the carbon content of dry matter accounts for approximately 45% of the total $NPP$ [56]. Therefore, the equation for calculating carbon sequestration ($CS$) is as follows:

$$CS = (NPP \div 0.45) \times 1.62 \tag{2}$$

Carbon Emissions

In previous studies, the total $CO_2$ emission data for China were usually calculated by referring to the method of calculating the pollutant emission coefficient described in the national Greenhouse Gas Inventory Guidelines published by the IPCC in 2006. This method has high practicality and applicability, and it has been widely used to calculate energy and fuel emissions [57–59]. In this paper, the method of Han et al. [22] was used. The specific formula are as follows:

$$AE = \sum_{m,n,t} (EC_{m,n,t} \times NCV_{m,n} \times EF_{m,n}) \tag{3}$$

where $m$, $n$, and $t$ are the industry, fuel type, and year of investigation, respectively; $AE$ is the carbon emissions (in metric tons), and $EC$ is the energy consumption (in metric tons). $NCV$ is the net calorific value (MWh/t) and $EF$ is the $CO_2$ emission factor (t $CO_2$/MWh). The details of $NCV$ and $EF$ refers to the study of Han et al. [22]. Due to the lack of energy consumption data for some counties in the YRD region, in our study, a particle swarm optimization–back propagation (PSO-BP) neural network model was used to simulate the carbon emissions of counties with missing data using the MATLAB platform with reference to Chen et al. [60]. To prevent overfitting, the MATLAB platform uses the method of dividing the data into three parts: training, validation, and testing. Only the training data were used in the training step, and the other two datasets that were not used in the training were used for testing. The collected carbon emission data for the counties were used as training data and testing data, and the counties with missing data were predicted and simulated. The training sample size was 70% of the output layer data in the different years. We achieved good simulation results (see Supplementary Figure S1). The PSO-BP neural network codes used in this study are shown in the Supplementary File (Code).

Carbon Budget

Zhao et al. [61] defined the regional carbon budget as the comparative relationship and balance between carbon sequestration and carbon emissions caused by all natural and man-made activities in a certain region in a specific period of time. Our study calculated the carbon budget with reference to Li et al. [62], and the formula is as follows:

$$CB = CS - AE \tag{4}$$

In Equation (4), $CB$ is the amount of carbon budget, $CS$ is the amount of carbon sequestration, and $AE$ is the amount of carbon emissions. If the value of $CB$ is positive, it means that the carbon budget is in surplus; if the value of $CB$ is negative, it means that the carbon budget is in deficit.

2.3.2. County-Level Data Analysis

Using the county level as the study unit is conducive to a more specific analysis of the temporal and spatial characteristics of regional carbon budgets and the formulation of more targeted emission reduction measures. Based on the Zonal Statistics tool of ArcGIS 10.8, our study counted and analyzed the values of the carbon budget and drivers at the county level. Then, we further divided the county-level statistical values into six categories to better demonstrate their evolution characteristics.

2.3.3. Driver Analysis

Referring to the research by Sun et al. [63] and after considering the natural status, development characteristics, and data availability of the YRD region, at the county scale, we selected 11 indicators as the potential drivers of carbon budget changes. The drivers included climate change (average temperature, average annual precipitation, and annual solar radiation per unit area) and human activities (proportion of cultivated land, forestland, grassland, water area, built-up area, unused land; population density; and per capita GDP). Changes in all potential drivers from 2000 to 2015 were first calculated and then regressed with changes in carbon budget per unit area.

After selecting the potential drivers, we first performed collinearity diagnosis based on the ordinary least squares (OLS) regression model and eliminated potential drivers with a variance inflation factor (VIF) greater than 7.5. Then, a Geodetector and geographically weighted regression (GWR) model were used to carry out global and local driving factor analysis. Geodetector can be used to detect spatial heterogeneity and find out the driving mechanism behind it [64]. The global impact of drivers on the carbon budget changes was analyzed by the factor detection tool in Geodetector. The GWR model is a local regression model based on the OLS model, which can reflect the degree of influence of different geographical variables on the region [65].

We used the OLS and GWR tools in ArcGIS 10.8 for the modeling. The Geodetector can be accessed at the website www.geodetector.cn (accessed on 10 November 2021).

## 3. Results

### 3.1. Changes in Carbon Sequestration and Emissions

From 2000 to 2015, both carbon sequestration and carbon emissions in the YRD region showed an increasing trend, where carbon sequestration increased by 4.18%, while carbon emissions increased by 215.86% (Table 2). Among the four provinces, Anhui and Jiangsu both increased carbon sequestration to some extent, while the other two showed a decreasing trend. Shanghai, in particular, saw a 14.51% reduction in carbon sequestration. In terms of carbon emissions, Jiangsu, Zhejiang, Shanghai and Anhui were ranked according to the increase in the total amount. According to the increase in carbon emissions per unit area, the rankings were Shanghai (433.78 t/ha), Jiangsu (64.20 t/ha), Zhejiang (32.16 t/ha) and Anhui (14.50 t/ha).

**Table 2.** Quantitative changes in carbon sequestration and emissions during 2000–2015.

| | Area (ha) | Carbon Sequestration (Mt) | | | Carbon Emission (Mt) | | |
|---|---|---|---|---|---|---|---|
| | | 2000 | 2015 | 2000–2015 | 2000 | 2015 | 2000–2015 |
| YRD region | $3.59 \times 10^7$ | 968.82 | 1009.36 | 40.54 | 697.49 | 2203.12 | 1505.63 |
| Anhui | $1.40 \times 10^7$ | 368.86 | 394.82 | 25.95 | 161.22 | 364.38 | 203.16 |
| Jiangsu | $1.07 \times 10^7$ | 225.53 | 243.19 | 17.67 | 286.39 | 974.57 | 688.18 |
| Zhejiang | $1.06 \times 10^7$ | 359.41 | 358.51 | −0.90 | 145.96 | 485.21 | 339.25 |
| Shanghai | $0.63 \times 10^6$ | 15.02 | 12.84 | −2.18 | 103.92 | 378.96 | 275.04 |

*3.2. Spatiotemporal Dynamics of the Carbon Budgets*

3.2.1. Changes in the Amount of Carbon Budget from 2000 to 2015

In 2000, the total carbon budgets of Anhui, Zhejiang and the entire study area were in surplus, while Jiangsu and Shanghai were in deficit (Figure 3a). Specifically, the ranking of carbon budgets was as follows: Zhejiang, Anhui, Jiangsu and Shanghai, which were 213.45, 207.65, −60.86 and −88.90 million tons, respectively. In 2015, only Anhui's carbon budget was in surplus, and the other three provinces were in deficit. Jiangsu had the largest carbon budget deficit gaps (−731.37 million tons), followed by Shanghai (−366.12 million tons) and Zhejiang (−126.70 million tons). In terms of changes in carbon budgets from 2000 to 2015, all four provinces showed a decreasing trend. The order of carbon budget reduction was Jiangsu, Zhejiang, Shanghai and Anhui. From 2000 to 2015, the carbon budget of the entire YRD region decreased by 1465.09 million tons.

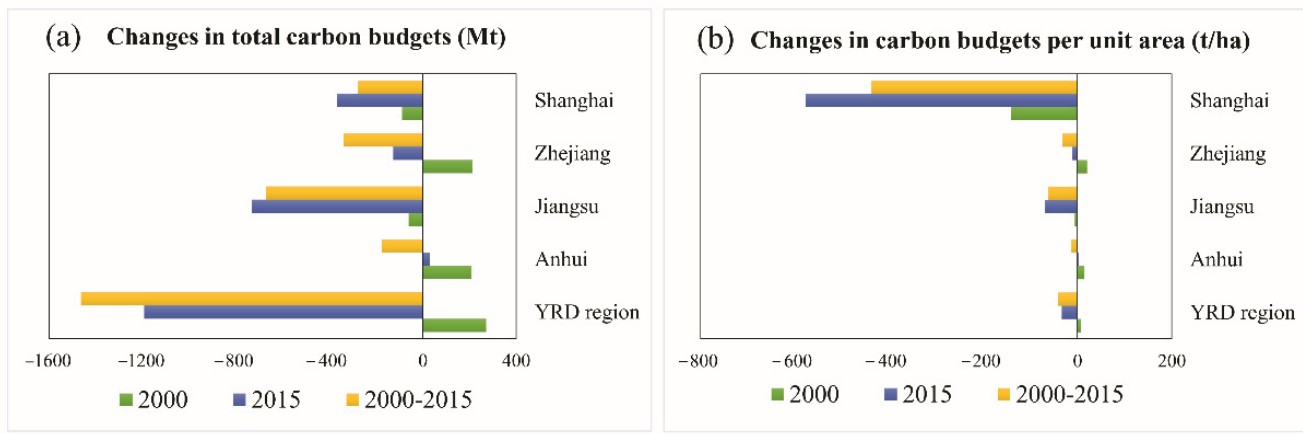

**Figure 3.** Quantitative changes in carbon budgets during 2000–2015.

In 2000, the carbon budget per unit area was negative in Shanghai, Jiangsu, and the YRD region and positive in Anhui and Zhejiang (Figure 3b). In 2015, only Anhui's carbon budget per unit area was positive, and the other three provinces were all negative, especially Shanghai, which reached −577.43 t/ha. From 2000 to 2015, the changes in the carbon budget per unit area in the four provinces all showed a decreasing trend, among which Shanghai decreased the most (−437.21 t/ha), followed by Jiangsu (−62.55 t/ha), Zhejiang (−32.24 t/ha) and Anhui (−12.65 t/ha).

3.2.2. Spatial Heterogeneity of the Carbon Budget Changes

We investigated the changes in the total carbon budget (Figure 4) and in the carbon budget per unit area (Figure 5) of each county in the YRD region from 2000 to 2015. In 2000, the carbon budget of most counties in the study area was in surplus, while the counties in deficit were mainly distributed in southern Jiangsu, northern Zhejiang and Shanghai (Figure 4a; Figure 5a). In 2015, more counties went from surplus to deficit in the carbon budget, and deficit gaps were increasing in many areas (Figure 4b; Figure 5b). From the results of carbon budget changes, most counties showed a decreasing trend from 2000 to 2015, and the areas with a large decrease were located in Shanghai, southern Jiangsu and northern Zhejiang; only a few counties showed an increasing trend, which were scattered in northern Jiangsu, southern Zhejiang and northern Anhui (Figure 4c; Figure 5c).

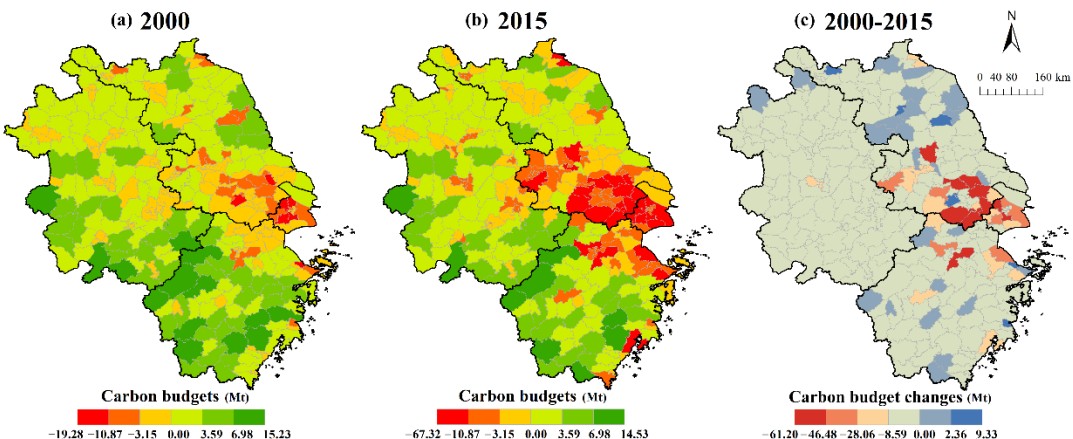

**Figure 4.** Spatial distribution and changes in the carbon budget from 2000 to 2015 in the YRD region.

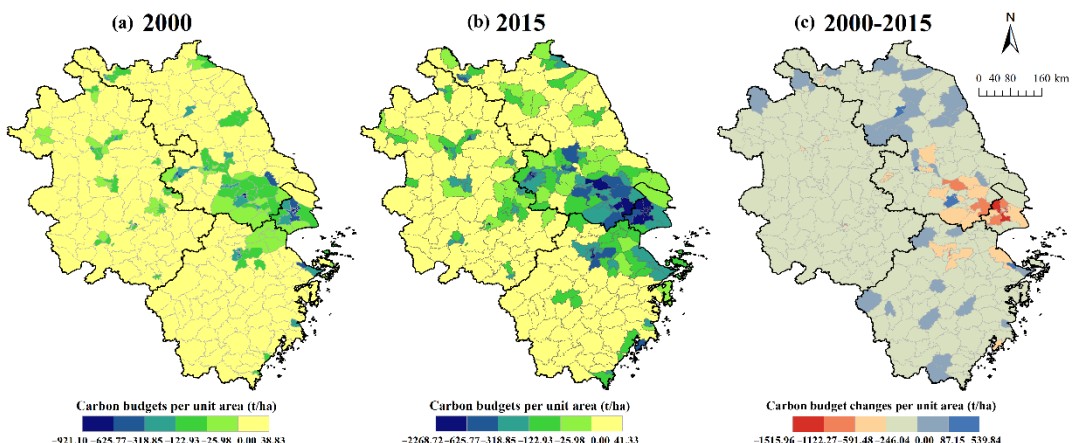

**Figure 5.** Spatial distribution and changes in the carbon budget per unit area from 2000 to 2015 in the YRD region.

### *3.3. Driver Analysis*

By using the carbon budget changes in the YRD region as the dependent variable and taking climate change (average temperature, average annual precipitation, and annual solar radiation per unit area) and human activities (proportion of cultivated land, forestland, grassland, water area, built-up area, unused land; population density; and per capita GDP) as the independent variables, the OLS model was first constructed. According to the OLS results, the variance inflation coefficient (VIF) values of the proportion of cultivated land and built-up area exceeded 7.5. To avoid multicollinearity, these two factors were excluded, and the remaining nine factors were selected. The GWR tool in ArcGIS provides standard error coefficients that measure the reliability of each coefficient estimate. These estimates are more confident when the value of the standard error coefficient is relatively small. Spatial distributions of standard error coefficients for each driver can be found in the Supplementary File (Figure S2).

#### 3.3.1. Global Analysis

Our study analyzed the impact of each driver on the carbon budget changes based on the factor detection module in Geodetector. The magnitude of the q value represents the influence of each driver. The results showed that changes in population density had the greatest impact on the carbon budget from 2000 to 2015, followed by changes in GDP per capita, the proportion of unused land, and average temperature per unit area, with q values of 0.3317, 0.1202, 0.0998, and 0.0928, respectively (Figure 6). Population, GDP, and unused land changes are factors in human activities, and temperature changes are factors

in climate change. The q values of the remaining drivers were not high, but they could still reflect the differences in the impact of different factors on the carbon budget changes. Changes in the proportion of grassland and annual average precipitation had less effect on the carbon budget.

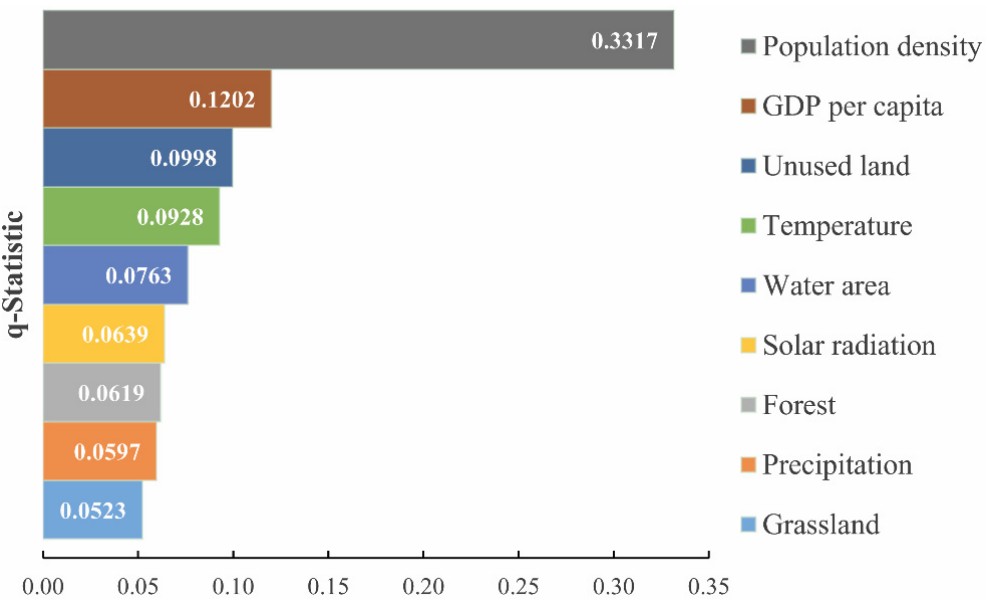

**Figure 6.** Impact of various drivers on carbon budget changes based on Geodetector analysis. All factors passed the significance test with *p* values less than 0.01. The land use type refers to the proportion of the total area, and the climate change factor refers to the quantity value per unit area.

### 3.3.2. Local Analysis

The results of GWR analysis showed that there is obvious spatial heterogeneity in the impact of various driving factors on carbon budget changes (Figure 7). Temperature and carbon budget changes were negatively correlated in southern Anhui and southern Jiangsu (around the Yangtze River) and were positively correlated in other regions. The areas with positive precipitation coefficients were mainly located in the east and south of Jiangsu, and the other areas in the YRD region had negative values. The areas with negative solar radiation coefficients were mainly located along the eastern coast of the YRD region, while most of the western regions were positive. The forest coefficient showed the characteristics of a circle. The coefficient value centered on Shanghai was positive and the largest, and the farther away from the center, the smaller the coefficient value which gradually became negative. The influence of grassland on carbon budget changes also had the characteristics of a circle, but it had become centered in eastern Zhejiang, and the evolution trend in coefficient values was opposite to that of forest. The spatial distribution of water area coefficients was similar to that of solar radiation, with negative values mainly distributed in the eastern coastal zone of the study area and positive values in other regions. The coefficients of unused land were high in the southeast and low in the northwest. Except for some counties in the western part of Anhui, where the coefficients were negative, all other regions were positive. Population density was negatively correlated with carbon budget changes in the entire study area, and the minimum coefficient values were located in the western fringe counties of Anhui and the junctions of Jiangsu, Zhejiang and Shanghai. The per capita GDP coefficients were positive in a few counties in eastern Zhejiang and negative in other areas; the minimum values were located in the middle of the YRD region.

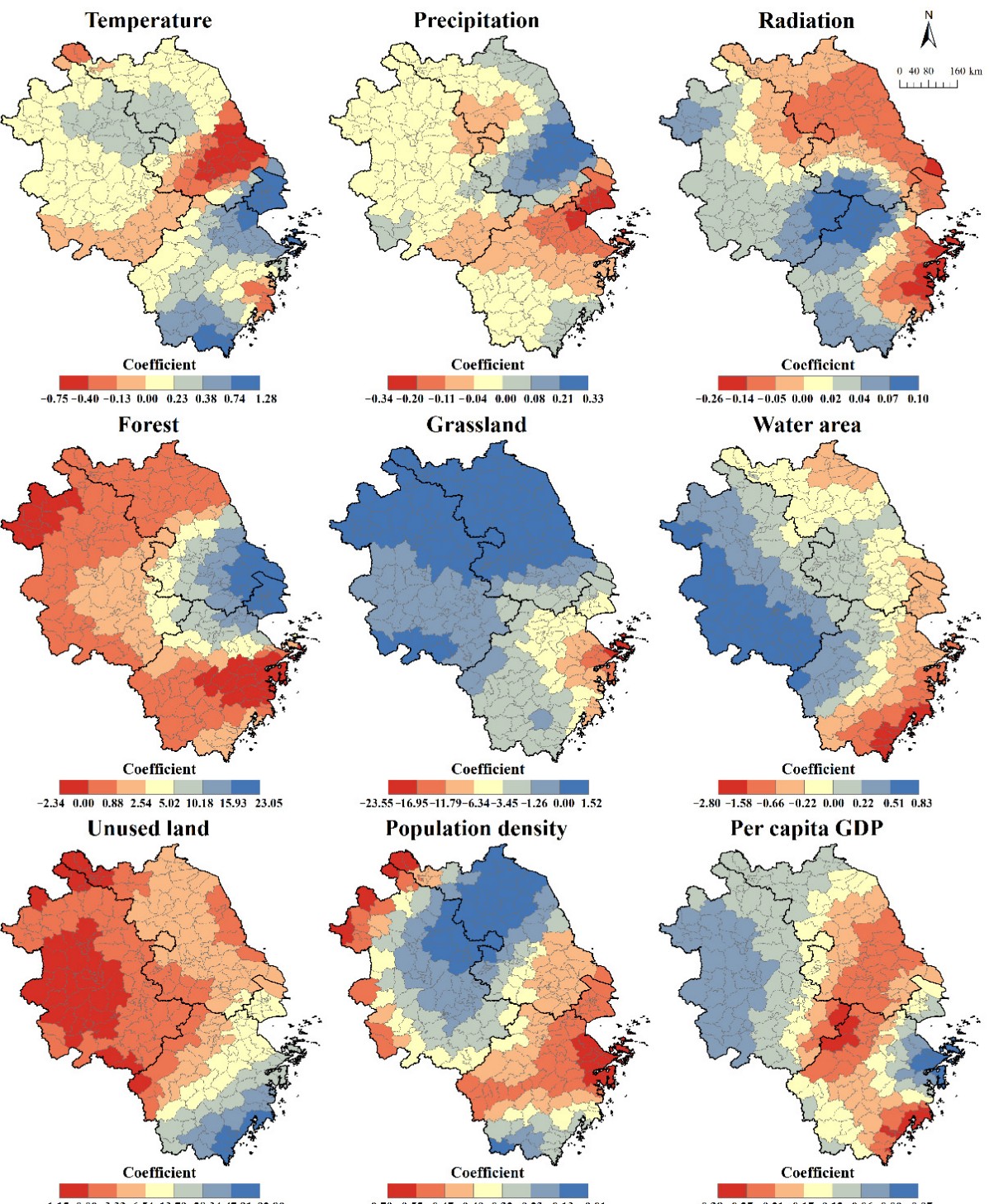

**Figure 7.** Spatial distribution of the GWR coefficients between carbon budget changes and each driver in the YRD region. A positive coefficient indicates a positive correlation between the driver and the carbon budget changes, and a negative coefficient indicates a negative correlation.

## 4. Discussion

In this paper, we combined the remote-sensing model and the IPCC inventory method to explore the spatiotemporal pattern changes in the carbon budget in the YRD region of China and analyzed the drivers leading to the changes. Different from previous studies, this study focused on county-scale analysis and analyzed the comprehensive effects of multiple drivers on carbon budgets. The issues we tried to discuss were threefold: (1) to

determine the characteristics of the spatiotemporal pattern of the carbon budget in the YRD region, (2) to analyze the driving effect of different factors on carbon budget changes, and (3) to discuss the implications, limitations and future perspectives of this study.

*4.1. Spatiotemporal Patterns of Carbon Budgets*

Our study found that the increase in carbon sequestration in the YRD region from 2000 to 2015 was much smaller than the increase in carbon emissions, and the carbon budget deficit areas were mainly located in Shanghai, southern Jiangsu, and northern Zhejiang, which is consistent with the findings of Tao et al. [53]. The rapid urbanization process, the increase in impervious surface area, and the large energy consumption in these areas were the main reasons for this phenomenon. As the core city of the YRD region, Shanghai is the growth pole of regional development. Many studies have shown that the socioeconomic development of the YRD region has a "core-edge" structure [66–68]. We found that the spatial pattern characteristics of carbon budgets in the YRD region are as follows: with Shanghai as the center, it gradually changes from a carbon budget deficit to a surplus as the distance increases, and the closer to Shanghai, the greater the deficit. This indicated that the spatial pattern of the carbon budget in the YRD region also has a "core-edge" structural feature. Although few studies have focused on the carbon budget in the YRD region, the results of some carbon sequestration or carbon emission studies conducted in this region can also support our speculation. For example, the research conducted by Sun et al. in the YRD region showed that the ecosystem carbon storage gradually increased with the distance from Shanghai [69]; Han et al. showed that the closer to Shanghai, the greater the value of net carbon emissions (the difference between carbon sources and carbon sinks in each pixel) [22]; Liu et al. demonstrated that the spatial network of the carbon emission efficiency in the YRD region had a core–edge structure [70]. Similarly, the carbon emissions of the Pearl River Delta, [71,72], Los Angeles, Moscow, Tokyo, New Delhi, and Sao Paulo also have this spatial pattern [73]. According to the first law of geography, the correlation between objects is related to the distance. Generally, the closer the distance is, the greater the correlation between objects, and vice versa. The carbon budget is also applicable to this law; that is, adjacent areas have similar pattern characteristics due to their similar dominant functional positioning. Shanghai and its surrounding areas have a high degree of geographical connection, and the dominant function is economic development. The large amount of energy consumption and the limited area of natural ecosystems make the region's carbon budget deficit serious.

*4.2. Drivers of Carbon Budget Changes*

The factors influencing the carbon budget of terrestrial ecosystems vary in different regions [74]. In this study, changes in population density, per capita GDP, and unused land area accounted for the top three impacts on the carbon budget in the YRD region (Figure 6). This shows that compared with climate change factors, human activities have a greater impact on the carbon budget in the YRD region. Moreover, we found that the effects of different drivers on the spatiotemporal pattern of the carbon budget were different (Figure 7). The results indicated that changes in population density have a much larger impact on the carbon budget than other drivers and were negatively correlated across all regions. This is consistent with Tan et al. [37], who concluded that population growth has a positive effect on China's carbon emissions. For the core area, an increase in population means an increase in food consumption, housing needs and transportation needs, resulting in more energy consumption and carbon emissions [75]. For the peripheral area, the primary industry practitioners continue to flow to the secondary and tertiary industries of the core area or local [76]. The reduction in the labor force in the primary industry means an enhancement of mechanization; the development of the secondary industry means an increase in energy consumption. These will all increase carbon emissions. Our study showed that an increase in per capita GDP in the YRD region leads to a reduction in the carbon budget. On the one hand, industrial development needs to obtain a large quantity

of raw materials, energy and resources from the region for the production process, resulting in carbon emissions; on the other hand, the construction of factories occupies forestland, grassland and wetlands, resulting in reduced carbon sequestration [77]. The increase in population and GDP in the YRD region is also accompanied by the development of much unused land into construction land, such as roads and high-rise buildings. These urban infrastructures and high-rise buildings will increase the demand for energy-intensive raw materials (such as steel, cement, etc.), thereby increasing carbon emissions [78]. In addition, a large increase in impervious surfaces comes at the cost of encroaching on urban public space, which will weaken the carbon sequestration function of natural ecosystems.

The relationship between water area and carbon budget is negatively correlated in the east of the study area and positively correlated in the west, which is related to the imbalance of economic development in the east and west of the YRD region. The eastern coastal areas have a high degree of urbanization, and the construction and development of cities pay more attention to the quality of life of residents. In recent years, many blue infrastructures, such as lakes and wetlands, have been added to cities in the eastern region [79]. However, most of the increased water was converted from woodland, grassland or cultivated land [80], thus leading to a decline in carbon sequestration and, in turn, a reduction in the carbon budget. Due to the relatively backward economy in the western part of the study area, the local government still regards economic development as the primary task. This has resulted in many water areas being landfilled and developed for construction, reducing the carbon budget. Forests are important carbon sinks in terrestrial ecosystems [34,81]. In this study, forest and carbon budgets were positively correlated in most areas with significant spatial heterogeneity. The closer to the core area, the greater the impact of forest change on the carbon budget. This shows that compared with peripheral cities, forests in core areas play a more critical role in regulating the carbon budget. Grassland accounts for a small proportion of land use types in the YRD region and has little impact on the regional carbon budget as a whole. Grassland is mainly distributed in the southern area, and a large area of grassland was converted into forest during the study period [49,82]. The cultivated land is mainly distributed in the northern area. Due to the Grain for Green program, some cultivated land was converted into grassland. Therefore, the relationship between grassland and carbon budget changes was negatively correlated in the southern part of the YRD region and positively correlated in the northern part.

In terms of climate change factors, the overall impact of temperature on the carbon budget is greater than that of solar radiation and precipitation (Figure 6), which is consistent with the study by Zhang et al. [83]. We found that the effect of temperature on the carbon budget is the opposite to that of precipitation in many regions. For example, temperature and carbon budget are negatively correlated in southern Jiangsu, while precipitation and carbon budget are positively correlated; temperature and carbon budget are positively correlated in the northern area, Shanghai and most of Zhejiang Province, while precipitation and carbon budget are negatively correlated. Durand et al. demonstrated that increasing solar radiation can promote photosynthesis in vegetation, thereby increasing carbon sequestration [84]. It may also exacerbate the urban heat island effect and lead to an increase in carbon emissions [85]. In our study, the effect of radiation on the carbon budget showed an opposite situation in the east and the west. This may be due to the high degree of urbanization in the eastern region. High-density buildings, asphalt pavements, and cement pavements have larger heat absorption rates and smaller specific heat capacities, which will further enhance the urban heat island effect under the influence of increased radiation [86]. To mitigate the heat island effect, the eastern region consumes more energy to cool down. The increase in carbon sequestration caused by increased radiation is not enough to offset this part of carbon emissions. Due to the low degree of urbanization in the western area, the impact of radiation on carbon sequestration is greater than that of carbon emissions. Therefore, radiation in these regions is positively correlated with the carbon budget.

*4.3. Implications, Limitations and Future Perspectives*

Our study found that the spatial pattern of the carbon budget in the YRD region has a "core-edge" structural feature. As the core of the YRD region, Shanghai and its surrounding areas have high population density and rapid economic development and are under the pressure of an unbalanced carbon budget. The peripheral area farther from the core area assumed the main carbon sink function. Furthermore, we revealed that population density, GDP per capita, and unused land change have the largest impacts on the carbon budget. Low-carbon development not only requires effectively curbing carbon emissions from the perspective of carbon sources but also requires increasing carbon sinks as much as possible, thereby improving the level of carbon sequestration in the region. Based on these study results, we make recommendations from three aspects: population, industrial development, and land use. (1) Limit the continuous migration of the population from the peripheral area to the core area. The increase in the urban population will inevitably bring about a larger gathering of economic activities and an increase in built-up areas, which in turn will result in more energy consumption and carbon emissions. Therefore, this is a key factor in regulating the regional carbon budget. (2) Upgrade high-energy-consuming industries in the core area and further promote regional economic growth and low-carbon development through innovation; continue to carry out afforestation projects in peripheral areas and develop ecological industries. Based on the accounting results of the carbon budget, the government can formulate ecological compensation measures and improve the carbon-trading market to compensate for the carbon sink functional area. In this way, on the one hand, the counties in the peripheral area can use this part of the funds to improve the salaries, employment opportunities, and medical and education levels of residents to retain the local labor force and reduce the population pressure in the core area. On the other hand, the counties of peripheral area can also invest more funds in ecological protection projects. To support policy formulation, our research provided a list of counties with severe reductions and counties with increases in carbon budgets in the YRD region from 2000 to 2015 (Supplementary Tables S1 and S2). (3) Implement a compact and three-dimensional urban space development model in the core area to promote the intensive use of land, control the excessive growth of artificial surfaces in the peripheral area and reduce the occupation of natural open space.

Our results provide a scientific basis for local green and low-carbon development policies. However, this study still has some limitations that need to be improved. For example, the carbon budget of terrestrial ecosystems is scale-dependent. This study assumed that the YRD region is a closed area and did not consider its carbon cycle relationship with the larger external terrestrial ecosystem. Due to data availability limitations, soil carbon storage was not considered in this paper, which may lead to an underestimation of total carbon sequestration. In addition, policy formulation also has an impact on the carbon budget changes, but this study did not involve policy scenario simulation.

Despite the above limitations, the method used in this paper is highly maneuverable and can quickly and effectively reveal the spatiotemporal patterns of the regional carbon budget and its drivers. In the future, we plan to work on the following aspects to further improve the current research. First, we will conduct sampling of vegetation and soil quadrats to verify the accuracy of the CASA model and supplement the soil carbon storage component. Second, we will combine global carbon dioxide satellite-monitoring data with the IPCC inventory approach to improve the simulation accuracy of carbon emissions. Third, we plan to search for available regional policy data and analyze the impacts of policy factors on carbon budget changes.

## 5. Conclusions

This study explored the spatiotemporal pattern characteristics of the carbon budget and analyzed its drivers in the YRD region. The results showed that both carbon sequestration and carbon emissions in the study area increased from 2000 to 2015, but the increase in carbon emissions far exceeded the carbon sequestration. The study area changed from a

carbon balance surplus in 2000 to a deficit in 2015. From the perspective of spatial patterns, the carbon budget in the YRD region has a "core-edge" structural feature. The closer it is to Shanghai, the core area, the more severe the carbon budget deficit; the farther from it, the greater the carbon budget surplus. We found that population density, GDP per capita and unused land change accounted for the top three impacts on the carbon budget in the YRD region. Overall, human activity has a larger impact on the carbon budget than climate change, and these impacts are mostly negative. Locally, the impact of each driver on the carbon budget showed obvious spatial heterogeneity. The different natural environments and socioeconomic development in different regions are the reasons for this spatial heterogeneity. Based on the results, we proposed limiting the continuous migration of the population from the peripheral area to the core area, carrying out industrial restructuring in the core area, and focusing on implementing ecological protection projects in the peripheral area, implementing a compact and three-dimensional urban space development model in the core area, and controlling the occupation of open natural spaces by built-up land in peripheral areas. Our study can provide a scientific basis for low-carbon development in the YRD region. Our methodology and findings can provide references for similar studies in other urbanized regions around the world.

**Supplementary Materials:** The following supporting information can be downloaded at: https://www.mdpi.com/article/10.3390/land11081230/s1, Figure S1: training and testing results of the relationship between carbon emissions and the sum of the Digital Number (DN) values of the counties in the YRD (a) 2000 and (b) 2015; Figure S2: the standard error coefficient of each driver; Table S1: counties with reductions in carbon budgets; Table S2: counties with increases in carbon budgets; Codes: codes used in Matlab for realizing the PSO-BP neural network model.

**Author Contributions:** Conceptualization, Q.F., X.B. and J.C.; methodology, Q.F. and M.G.; software, M.G.; formal analysis, Q.F.; resources, Y.W. and T.W.; data curation, M.G.; writing—original draft preparation, Q.F. and M.G.; writing—review and editing, Q.F. and X.B.; visualization, Q.F.; supervision, J.C.; project administration, Q.F.; funding acquisition, Q.F. All authors have read and agreed to the published version of the manuscript.

**Funding:** This research was funded by the National Natural Science Foundation of China (42101253), the Jiangsu Social Science Foundation (19GLC016), the Major Projects of Philosophical and Social Sciences Research in Colleges and Universities in Jiangsu Province (2019SJZDA043), and the Open fund of State Key Laboratory of urban and regional ecology (SKLURE2021-2-2).

**Data Availability Statement:** All data generated or analyzed during this study are included in this published article.

**Acknowledgments:** We would like to thank the anonymous reviewers for their valuable comments and suggestions.

**Conflicts of Interest:** The authors declare no conflict of interest.

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
