# Peer review of "Spatiotemporal Patterns and Drivers of the Carbon Budget in the Yangtze River Delta Region, China"

_land, doi:10.3390/land11081230_

Round 1

Reviewer 1 Report

I carefully analyzed the paper. At this moment it is necessary to revise the authors. The paper is not written in an academic way and does not present the accuracy necessary for a scientific paper. However, I consider that the paper has the potential to contribute to the development of knowledge and I recommend improving it.

The following improvements are needed:

1. Improve the accuracy of the paper, follow the instructions of the sustainability journal for editing.

2. Redo the abstract so as to include information about the results obtained.

3. Improve the documentation of the approached topic, specify the research hypothesis and specify the purpose and objectives of your research.

4. Specify the references used for the methodology used.

5. Improve the references in the discussion chapter.

Author Response

Dear Editor and Reviewers,

We are very grateful to you for reviewing our manuscript and providing detailed comments. Your comments and suggestions are very helpful for revising and improving our paper, as well as the important guiding significance to our research. We have studied comments carefully and have made correction which we hope meet with approval. “Track Changes” function was used and revised portion were marked in red in the paper. The page and line numbers in parentheses below correspond to the version of manuscript with track changes. The main corrections in the paper and the responses to the reviewer’s comments are as flowing:

Reviewer 1:

I carefully analyzed the paper. At this moment it is necessary to revise the authors. The paper is not written in an academic way and does not present the accuracy necessary for a scientific paper. However, I consider that the paper has the potential to contribute to the development of knowledge and I recommend improving it.

The following improvements are needed:

Comment 1: Improve the accuracy of the paper, follow the instructions of the sustainability journal for editing.

Response to comment 1: Thank you for this comment. We have carefully checked and edited the format of the paper in accordance with the journal's instructions.

Comment 2: Redo the abstract so as to include information about the results obtained.

Response to comment 2: Done accordingly (Page 1, line 16-34).

Comment 3: Improve the documentation of the approached topic, specify the research hypothesis and specify the purpose and objectives of your research.

Response to comment 3: Thank you very much for your suggestions. We have revised the Introduction to make the research goals and contributions more explicit (Page 3, line 110-123).

Comment 4: Specify the references used for the methodology used.

Response to comment 4: We totally agree with you. We have specified the references in the Method part (Page5, line 174; Page 7, line 212 and 239; Page 8, line 254, 268 and 271) and shortened the textual description of the method to make it more concise. In addition, we have added a flow chart based on the suggestions of other reviewers (Page 5, line 154-164).

Comment 5: Improve the references in the discussion chapter.

Response to comment 5: Many thanks for your comment. We have already made revisions in the Discussion chapter (Page 15, line 450, 454; Page 16, line 470, 474, 508, 513 and 518).

We tried our best to improve the manuscript and made some changes in the manuscript. These changes will not influence the content and framework of the paper. We appreciate for your warm work earnestly and hope that the correction will meet with approval. Once again, thank you very much for your comments and suggestions.

Best regards,

Qi Fu, Xu Bi, and Jinhua Chen

Reviewer 2 Report

I am very interested in the manuscript you submitted over. Carbon emissions, carbon sequestration and carbon budgets are hot issues discussed in the international community in recent years. The study of regional carbon budgets is crucial to mitigating climate change and achieving low-carbon development. Although there is certain research base on carbon emissions and carbon sequestration, there is relatively little discussion on regional carbon budgets, especially in the rapidly urbanizing regions.

As the largest urban agglomeration in eastern China, the Yangtze River Delta is experiencing rapid urbanization, the imbalance of regional development is becoming more and more prominent, and differentiated urban development strategies are imminent. Therefore, the results present a significant advance for a scientific question such as carbon budget. This is a careful study, but some parts need improvement before considering publication. The comments are as follows:

1. In the Introduction, the novelty needs to be clearly addressed. It may be better to add a detailed description of the environmental impact of the rapid urbanization of carbon in the Yangtze River Delta.

2. In 2.1 Study Area, we suggest introducing relations to the topic of this article, carbon sequestration and carbon budget, in order to let the readers know more about the region and the background about the region related to carbon.

3. In 2.2 Data sources, the data used in this manuscript is in 2015, which is relatively old. It would better be updated to 2022.

4. In 2.3.2 Carbon emissions, the format after Formula 6 is not standardized. Please revise it accordingly.

5. In 2.5 Driver analysis, the authors said that eleven indicators are selected to establish the index system, but didn’t clarify why these factors are selected. Please justify and provide relevant references.

6. More Policy recommendations for the county level should be added to the discussion section.

7. To show your method more clearly, it may be better to add a technical route figure.

8. Minimize the number of "in this study" in the whole manuscript.

9. The language needs to improve. Some expression needs to be concise. It is suggested to edit the manuscript carefully and thoroughly by a native English speaker, to make it clear and easier for readers to understand.

Author Response

Dear Editor and Reviewers,

We are very grateful to you for reviewing our manuscript and providing detailed comments. Your comments and suggestions are very helpful for revising and improving our paper, as well as the important guiding significance to our research. We have studied comments carefully and have made correction which we hope meet with approval. “Track Changes” function was used and revised portion were marked in red in the paper. The page and line numbers in parentheses below correspond to the version of manuscript with track changes. The main corrections in the paper and the responses to the reviewer’s comments are as flowing:

Reviewer 2:

I am very interested in the manuscript you submitted over. Carbon emissions, carbon sequestration and carbon budgets are hot issues discussed in the international community in recent years. The study of regional carbon budgets is crucial to mitigating climate change and achieving low-carbon development. Although there is certain research base on carbon emissions and carbon sequestration, there is relatively little discussion on regional carbon budgets, especially in the rapidly urbanizing regions.

As the largest urban agglomeration in eastern China, the Yangtze River Delta is experiencing rapid urbanization, the imbalance of regional development is becoming more and more prominent, and differentiated urban development strategies are imminent. Therefore, the results present a significant advance for a scientific question such as carbon budget. This is a careful study, but some parts need improvement before considering publication. The comments are as follows:

Comment 1: In the Introduction, the novelty needs to be clearly addressed. It may be better to add a detailed description of the environmental impact of the rapid urbanization of carbon in the Yangtze River Delta.

Response to comment 1: Thank you for this comment. We have highlighted the novelty of the paper in the Introduction (Page 3, line 110-112). We have changed the sentence “but the large amount of carbon emissions has also caused significant harm to the environment” to “but it has also caused a lot of carbon emissions” (Page 3, line 101). Meanwhile, we added the impact of urbanization on carbon sequestration and carbon emissions in the YRD region in the Study Area section (Page 3, line 137-143).

Comment 2: In 2.1 Study Area, we suggest introducing relations to the topic of this article, carbon sequestration and carbon budget, in order to let the readers know more about the region and the

background about the region related to carbon.

Response to comment 2: Thank you for this suggestion. We have added the background about the carbon sequestration and emissions of the YRD region (Page 3, line 137-143).

Comment 3: In 2.2 Data sources, the data used in this manuscript is in 2015, which is relatively old. It would better be updated to2022.

Response to comment 3: Thank you for your suggestion. In fact we formed this research program and started collecting the necessary data starting in early 2019. At the beginning we originally intended to choose the study years as 2000 and 2018. But our data collection struggled because many counties had not released their 2018 energy statistics (for carbon emissions calculations) at that time. After careful consideration, we finally chose 2000 and 2015 as time periods for the study. Your advice is invaluable, and we'll focus on the near-term period in our upcoming research.

Comment 4: In 2.3.2 Carbon emissions, the format after Formula 6 is not standardized. Please revise it accordingly.

Response to comment 4: Thank you for your correction. We have revised it accordingly (Page 7, line 216).

Comment 5: In 2.5 Driver analysis, the authors said that eleven indicators are selected to establish the index system, but didn’t clarify why these factors are selected. Please justify and provide relevant references.

Response to comment 5: Thank you for this comment. Through previous literature reading, we found that the main influencing factors of carbon budget can be divided into two categories: climate change and human activities. Considering the natural status, development characteristics and data availability of the YRD region, and focusing on the research of Sun et al., we selected 11 potential influencing factors mentioned in the paper. We have added this reference on Page 8, line 254.

Sun, X.; Tang, H.; Yang, P.; Hu, G.; Liu, Z.; Wu, J. Spatiotemporal Patterns and Drivers of Ecosystem Service Supply and Demand across the Conterminous United States: A Multiscale Analysis. Science of the Total Environment 2020, 703, doi:10.1016/j.scitotenv.2019.135005.

Comment 6: More Policy recommendations for the county level should be added to the discussion section.

Response to comment 6: We completely agree with you. We have added the related sentence in the discussion section (Page 16, line 549-551). To support policy formulation, we provided a list of counties with severe reductions and counties with increases in carbon budgets in the Supplementary Table A1 and A2.

Comment 7: To show your method more clearly, it may be better to add a technical route figure.

Response to comment 7: We appreciate this comment. We have added it on Page 5, line 154-164.

Comment 8: Minimize the number of "in this study" in the whole manuscript.

Response to comment 8: Done accordingly.

Comment 9: The language needs to improve. Some expression needs to be concise. It is suggested to edit the manuscript carefully and thoroughly by a native English speaker, to make it clear and easier for readers to understand.

Response to comment 9: Thank you for your suggestion. Some sentences have been deleted or shortened. Native English speakers have helped us polish the paper.

We tried our best to improve the manuscript and made some changes in the manuscript. These changes will not influence the content and framework of the paper. We appreciate for your warm work earnestly and hope that the correction will meet with approval. Once again, thank you very much for your comments and suggestions.

Best regards,

Qi Fu, Xu Bi, and Jinhua Chen

Reviewer 3 Report

Manuscript entitled “Spatiotemporal patterns and drivers of the carbon budget in the Yangtze River Delta region, China” by Fu et al.,

Authors highlighted the importance of understanding of the patterns and drivers of regional carbon budgets.

Author reported based on the Carnegie-Ames-Stanford Approach (CASA) model, the IPCC inventory method, the Ge-odetector model, and the geographically weighted regression (GWR) method.

They have highlighted results. The carbon budget in the YRD region changed from a surplus in 2000 to a deficit in 2015. In terms of spatial pattern, the carbon budget of the YRD region has a “core-edge” structural feature. The top three drivers were, in order, changes in population density, GDP per capita, and unused land. Locally, the impact of the drivers on the carbon budget shows obvious spatial heterogeneity.

Abstract: overall excellent, but needs further refinement, such as what the authors consider the contribution of this paper.

Keywords: Ok

Introduction: Excellent, add latest relevant references.such as:
https://doi.org/10.1016/j.resconrec.2022.106411, https://doi.org/10.1007/s11356-021-134441, https://doi.org/10.1016/j.jclepro.2022.130966

MATERIAL AND METHODS: overall excellent (if possible, please add flow diagram of complete study.)

Data sources: OK

Figure 1 a. – The Ministry of Natural Resources of China requires that maps involving China must be used with maps containing review numbers. Of course, this is not mandatory for submission to an English journal. The authors are invited to consider this.

Figures 2. 3. – Good

The OLS and GWR: OK, but how did the author consider the significance test of the coefficients of influence factors when using the GWR model? Are all influencing factors significant?

Results: Good.

Figures 4. 5. 6: Excellent

Discussions: Entire discussion is very generalized, need major modification with incorporated latest relevant studies.

Conclusions: Good

Conclusions follow from the results and are reasonable. The article will be of interest to a wide range of readers whose scientific interests are related to ecology, environment, as well as climate change. Despite the fact that English is not my native language, I read the paper with interest and had no difficulties in understanding. The paper fully corresponds to the subject and level of the Land.

Author Response

Dear Editor and Reviewers,

We are very grateful to you for reviewing our manuscript and providing detailed comments. Your comments and suggestions are very helpful for revising and improving our paper, as well as the important guiding significance to our research. We have studied comments carefully and have made correction which we hope meet with approval. “Track Changes” function was used and revised portion were marked in red in the paper. The page and line numbers in parentheses below correspond to the version of manuscript with track changes. The main corrections in the paper and the responses to the reviewer’s comments are as flowing:

Reviewer 3:

Manuscript entitled “Spatiotemporal patterns and drivers of the carbon budget in the Yangtze River Delta region, China” by Fu et al., Authors highlighted the importance of understanding of the patterns and drivers of regional carbon budgets. Author reported based on the Carnegie-Ames-Stanford Approach (CASA) model, the IPCC inventory method, the Geodetector model, and the geographically weighted regression (GWR) method.

They have highlighted results. The carbon budget in the YRD region changed from a surplus in 2000 to a deficit in 2015. In terms of spatial pattern, the carbon budget of the YRD region has a “core-edge” structural feature. The top three drivers were, in order, changes in population density, GDP per capita, and unused land. Locally, the impact of the drivers on the carbon budget shows obvious spatial heterogeneity.

Specific Comment:

Abstract: overall excellent, but needs further refinement, such as what the authors consider the contribution of this paper.

Response to this comment: Thank you for this comment. We have added relevant sentences to abstract to highlight the contribution of the paper (Page 1, line 16-17).

Keywords: Ok

Introduction: Excellent, add latest relevant references. such as:

https://doi.org/10.1016/j.resconrec.2022.106411,

https://doi.org/10.1007/s11356-021-134441,

https://doi.org/10.1016/j.jclepro.2022.130966

Response to this comment: Thank you for your suggestion. We have added these references in the Introduction.

MATERIAL AND METHODS: overall excellent (if possible, please add flow diagram of complete study.)

Response to this comment: We totally agree with you. We have added the flow diagram of this study on Page 5, line 154-164.

Data sources: OK

Figure 1 a. – The Ministry of Natural Resources of China requires that maps involving China must be used with maps containing review numbers. Of course, this is not mandatory for submission to an English journal. The authors are invited to consider this.

Response to this comment: Thank you for your suggestion. In this study, the Chinese border vector data were sourced from the National Geographic Information Center. We confirm that these data are authoritative.

Figures 2. 3. – Good

The OLS and GWR: OK, but how did the author consider the significance test of the coefficients of influence factors when using the GWR model? Are all influencing factors significant?

Response to this comment: Thanks for this question. We conducted the GWR analysis by using ArcGIS 10.8. The GWR tool in ArcGIS doesn’t provide significance test as the OLS model does. But it provides standard error coefficients that measure the reliability of each coefficient estimate. These estimates are more confident when the value of the standard error coefficient is relatively small. We have explained it in the paper (Page 11, line 350-354) and added a Figure A2 about the spatial distributions of standard error coefficients for each driver in the Supplementary File.

Results: Good.

Figures 4. 5. 6: Excellent

Discussions: Entire discussion is very generalized, need major modification with incorporated latest relevant studies.

Response to this comment: Thank you for your comment. We have already made revisions in the Discussion chapter (Page 15, line 450, 454; Page 16, line 470, 474, 508, 513 and 518).

Conclusions: Good

Conclusions follow from the results and are reasonable. The article will be of interest to a wide range of readers whose scientific interests are related to ecology, environment, as well as climate change. Despite the fact that English is not my native language, I read the paper with interest and had no difficulties in understanding. The paper fully corresponds to the subject and level of the Land.

Response to this comment: Thanks for your affirmation. Your valuable comments help us a lot.

We tried our best to improve the manuscript and made some changes in the manuscript. These changes will not influence the content and framework of the paper. We appreciate for your warm work earnestly and hope that the correction will meet with approval. Once again, thank you very much for your comments and suggestions.

Best regards,

Qi Fu, Xu Bi, and Jinhua Chen

Round 2

Reviewer 2 Report

The authors have addressed my concerns. This article is suitable for publication in its present form.

This manuscript is a resubmission of an earlier submission. The following is a list of the peer review reports and author responses from that submission.